# Development of Marker Recycling Systems for Sequential Genetic Manipulation in Marine-Derived Fungi *Spiromastix* sp. SCSIO F190 and *Aspergillus* sp. SCSIO SX7S7

**DOI:** 10.3390/jof9030302

**Published:** 2023-02-26

**Authors:** Yingying Chen, Jiafan Yang, Cunlei Cai, Junjie Shi, Yongxiang Song, Junying Ma, Jianhua Ju

**Affiliations:** 1CAS Key Laboratory of Tropical Marine Bio-Resources and Ecology, Guangdong Key Laboratory of Marine Materia Medica, RNAM Center for Marine Microbiology, South China Sea Institute of Oceanology, Chinese Academy of Sciences, 164 West Xingang Road, Guangzhou 510301, China; 2Southern Marine Science and Engineering Guangdong Laboratory (Guangzhou), No. 1119, Haibin Road, Nansha District, Guangzhou 511458, China; 3College of Oceanology, University of Chinese Academy of Sciences, Qingdao 266400, China

**Keywords:** *pyrG* marker, *amdS* marker, marine-derived fungi, genome editing, secondary metabolite

## Abstract

Marine-derived fungi are emerging as prolific workhorses of structurally novel natural products (NPs) with diverse bioactivities. However, the limitation of available selection markers hampers the exploration of cryptic NPs. Recyclable markers are therefore valuable assets in genetic engineering programs for awaking silent SM clusters. Here, both *pyrG* and *amdS-*based recyclable marker cassettes were established and successfully applied in marine-derived fungi *Aspergillus* sp. SCSIO SX7S7 and *Spiromastix* sp. SCSIO F190, respectively. Using *pyrG* recyclable marker, a markerless 7S7-∆*depH* strain with a simplified HPLC background was built by inactivating a polyketide synthase (PKS) gene *depH* and looping out the *pyrG* recyclable marker after *depH* deletion. Meanwhile, an *amdS* recyclable marker system was also developed to help strains that are difficult to use *pyrG* marker. By employing the *amdS* marker, a backbone gene *spm11* responsible for one major product of *Spiromastix* sp. SCSIO F190 was inactivated, and the *amdS* marker was excised after using, generating a relatively clean F190-∆*spm11* strain for further activation of novel NPs. The collection of two different recycle markers will guarantee flexible application in marine-derived fungi with different genetic backgrounds, enabling the exploitation of novel structures in various fungi species with different genome mining strategies.

## 1. Introduction

Marine-derived fungi have yielded an increasing number of structurally unique and biologically active natural products (polyketides, alkaloids, steroids, terpenoids, and peptides) [1,2,3]. The chemical diversity of the compounds generated from marine-derived fungi, along with the strains’ novelty, shows that this group of microeukaryotes are tremendous sources for the isolation of unusual bioactive natural products [4]. Nonetheless, most of the natural product biosynthetic gene clusters (BGCs) are either dormant or weakly expressed under laboratory conditions. The plethora of advanced tools for genome sequencing and mining enabled the discovery of new secondary metabolites which are “silent” and are not expressed under laboratory conditions. For example, our previous studies have shown that metabolic blockade-based genome mining via generating a “multiple-deletion” mutant is an efficient method for exploring novel secondary metabolites (SMs) [5,6]. Likewise, genetic manipulations of pathway-specific regulatory genes and global transcription regulators are alternative methods for activating cryptic gene clusters [7,8]. Despite of the strategy used, a crucial element common to all of these genetic manipulation techniques is the use of a selection marker that enables discrimination between transformed and untransformed cells. However, for most marine-derived fungi, only a few markers are useful due to intrinsic resistance to drugs, which precludes the ability to sequentially disrupt/activate multiple genes, especially when chassis cells with simplified HPLC backgroud are needed before exploring novel natural products. Therefore, it is helpful to establish marker recycling systems to resolve the limitation of selection markers.

Currently, *pyrG* or *amdS*-based marker recycling systems are the two widely used methods in multistep gene editing. The *pyrG* gene encodes orotidine-5-phosphate (OMP) decarboxylase that catalyzes the decarboxylation of orotidine-5′-monophosphate to form uridine monophosphate [9]. In filamentous fungi, mutants of *pyrG* cannot grow without exogenous uridine/uracil (uridine/uracil auxotrophs). Additionally, OMP decarboxylase could also convert 5-fluoroorotic acid (5-FOA) to fluoroorotidine monophosphate, which can be further converted into fluorodeoxyuridine, a toxic compound that inhibits DNA and RNA synthesis. Resistance to 5-FOA can be achieved through the inactivation of OMP decarboxylase [10]. Therefore, the *pyrG* gene can be used as a maker allowing auxotrophic selection and 5-FOA counterselection permitting unlimited rounds of genetic manipulation. The generation of 5-FOA-resistant *pyrG* mutants has been reported in the different genera of filamentous fungi including *Aspergillus* sp. [10,11,12,13], *Acremonium* sp. [14], *Talaromyces versatilis* [10], and *Trichoderma reesei* [15]. Nevertheless, when applied to marine fungal strains with limited available sequencing and genetic information, the *pyrG* recycling system can be a laborious, time-consuming, and inconclusive process. An excellent alternative to the *pyrG*-based counterselection is the *amdS* selection marker as it can be directly used in the strain without the requirement of the null mutant as the recipient strain [16]. The *amdS* gene from *A. nidulans*, encoding acetamidase (*amdS*), was first characterized in 1970 by Hynes and Pateman [17]. Acetamidase catalyzes acetamide into acetate and ammonium allowing the organism to utilize acetamide as a sole carbon or nitrogen source. Acetamidase also hydrolyses the toxic fluoroacetamide to produce fluoroacetate that can be further transformed into toxic fluoroacetyl CoA and fluorocitrate, leading to inhibition of the citric acid cycle [17,18,19]. Hence, *amdS* can serve as a bidirectional selectable marker and has been employed in several members of Ascomycota that grow poorly on acetamide medium [16,20,21,22]. Despite the successful use of *pyrG* and/or *amdS* as recycling makers in various fungal taxa, these two counterselectable markers are seldom utilized in genetic manipulation of marine fungi.

*Aspergillus* sp. SCSIO SX7S7 and *Spiromastix* sp. SCSIO F190 are different fungal species isolated from marine environments [23,24,25]. Several antimicrobial natural products (e.g.,spiromarmycin, spiromastol L, and nornidulin) have been identified from these two strains with high yielding [24,25,26]. Genome sequencing and antiSMASH analyses revealed that both strains possess various cryptic secondary metabolites that need to be further activated by genetic approaches. Due to the high abundance of constitutively produced compounds in these two strains, inactivation of the BGCs corresponding to these major products before activation of the silent SMs would simplify the isolation steps of novel natural products. Moreover, we expected that, in some cases, inhibiting the production of major secondary metabolites would free up subunits for incorporation into other NPs. Therefore, multiple rounds of genetic manipulation are required for sequential inactivation/activation of the secondary metabolite BGCs. Our previous study has established a CRISPR-Cas9-based gene editing vector (pBSKII-toCas9-*hph*-sgRNA) with a hygromycin B resistant maker for these two strains [23]. However, it is difficult to recycle the *hph* maker gene once it is integrated into the genome of the host strain, which limits its application in multistep genome editing.

In this study, we developed a platform including both *pyrG* and *amdS*-based marker recycling systems by replacing hygromycin B phosphotransferase (*hph*) expression element in pBSKII-toCas9-*hph*-sgRNA with direction repeats (DRs) flanked *pyrG* or *amdS* expression cassette. The two new constructed vectors named pBSKII-toCas9-*pyrG*-sgRNA and pBSKII-toCas9-*amdS*-sgRNA were successfully applied for gene disruption in *Aspergillus* sp. SCSIO SX7S7 and *Spiromastix* sp. SCSIO F190, respectively. In *Aspergillus* sp. SCSIO SX7S7, the tricyclic depsidones and bicyclic depsides (DEPs) biosynthesis cluster was inactivated by disrupting of a PKS gene (*depH*), and the *pyrG* marker was removed after deletion of the *depH* gene, generating a 7S7-∆*depH* strain with a relatively simple HPLC background compared to the wild type strain for further activating of cryptic NPs by sequential genetic editing. In *Spiromastix* sp. SCSIO F190, the production of one major product spiromarmycin was abolished via inactivation of the PKS gene *spm11* using the *amdS*-based marker recycling system. Subsequential removal of the *amdS* marker in the ∆*spm11* strain allowed multiple rounds of genetic manipulation for inducing the expression of silent NPs.

## 2. Materials and Methods

### 2.1. Strains, Plasmids, and Culture Conditions

All strains and plasmids used in this study are listed in Appendix A. Both *Spiromastix* sp. SCSIO F190 and *Aspergillus* sp. SCSIO SX7S7 are marine-derived as indicated in our previous study [23]. All fungi strains (Appendix A) were routinely maintained on potato dextrose agar (PDA) and in potato dextrose broth (PDB) at 28 °C for the sporulation and the protoplast preparation, respectively. A 5-FOA sensitivity assay was performed on a PDA medium with different concentrations of 5-FOA. The acetamide plate was prepared according to the Aspergillus Nitrogen-free Minimal (ANM) medium [27] recipe using acetamide instead of glucose and ammonium [28]. Removing of the *amdS* marker gene was conducted on an ANM plate supplementing with 10 mM FAA. *Escherichia coli* strain DH5α was cultured in the Luria–Bertani (LB) medium with 1 mg/mL ampicillin for plasmid DNA isolation. The pBluescript II SK plasmid (Appendix A) for generating *pyrG/amdS* expression cassette [28] was kindly provided by Prof. Wong, K.H. from the University of Macau. The pBSKII-toCas9-*hph* plasmid used for constructing pBSKII-toCas9-*pyrG* and pBSKII-toCas9-*amdS* plasmids was kindly provided by Prof. Dan Hu from Jinan University [29]. The pFC330 plasmid (Appendix A) containing the sgRNA expression sequence and *AfpyrG* coding sequence was generously offered by Prof. Yi Tang from the University of California, Los Angeles, CA, USA [30]. The pBSKII-toCas9-*pyrG* and pBSKII-toCas9-*amdS* vectors (Appendix A) were used to construct the Cas9 and sgRNA-expressing vector pBSKII-toCas9-*pyrG/amdS*-sgRNA in this study.

### 2.2. DNA Manipulation

The primer synthesis and DNA sequencing were performed by Qingke Biotech Co., Ltd. (Guangzhou, China) and Sangong Biotech Co., Ltd. (Shanghai, China) (Appendix A). The plasmid extraction and DNA purification were carried out with commercial kits (Omega Bio-Tech, Inc., Norcross, GA, USA), following the manufacturer’s protocol. The one step cloning of multiple DNA fragments was carried out using a pEASY^®^-Uni Seamless Cloning and Assembly Kit from TransGen Biotechnology Co., Ltd. (Beijing, China), following the manufacturer’s protocol. The restriction enzymes and other DNA modification reagents were purchased from New England Biolabs, Inc. (NEB, Hitchin, UK) and Takara Biotechnology Co., Ltd. (Dalian, China), following the manufacturer’s protocol. The DNA amplification was performed on a PCR thermal cycler from Applied Biosystems (Thermo Fisher, Shanghai, China) with either Taq DNA polymerase (TransGen, Beijing, China) or High-fidelity Fastpfu DNA polymerase (TransGen, Beijing, China), following the manufacturer’s protocol.

### 2.3. Plasmid Construction

The pBSKII-toCas9-*pyrG*-sgRNA and pBSKII-toCas9-*amdS*-sgRNA plasmids were constructed by three steps as follows. First, the *pyrG* and *amdS* genes were amplified from FC330 and *A. niduans* using primers *pyrG*-F1/R1 and *amdS*-F1/R1, respectively. Either of the *pyrG* and *amdS* sequences together with *A. nidulans* derived P*trpC* and T*trpC* fragments were added in *SmaI* digested pBluescript II SK plasmid by one step cloning kit, yielding pBSKII-*pyrG* or pBSKII-*amdS*. Second, the *pyrG* and *amdS* expression cassettes under the control of P*trpC* and T*trpC* were amplified from the pBSKII-*pyrG* and pBSKII-*amdS* plasmids using primers DR-P*trpc*-F and *PsiI*-T*trpc*-R. In addition, the 300 bp DR sequence was amplified from pBSKII-toCas9-*hph* plasmid. Then, the *pyrG* or *amdS* expression cassette along with the DR sequence were ligated into *PsiI* cut pBSKII-toCas9-hph plasmid to replace the *hph* expression cassette, generating pBSKII-toCas9-*pyrG* or pBSKII-toCas9-*amdS*. Third, the sgRNA for targeting gene together with ribozyme sequences controlled by the P*gdpA* and T*trpC* sequences was separately amplified from pFC330 using two primer pairs Frag1-F/R and Frag2-F/R. Next, the two sgRNA expression fragments were inserted into the *BsaAI* digested pBSKII-toCas9-*pyrG* or pBSKII-toCas9-*amdS* plasmid by one step cloning.

### 2.4. 5-FOA Sensitivity Assay

The 5-FOA sensitivity was determined in PDA plate with varying concentrations of 5-FOA. The spores of *Aspergillus* sp. SCSIO SX7S7 were inoculated on a PDA medium supplemented with 5-FOA at gradient concentrations (0, 0.5, 1, 2 mg/mL). Control plates without 5-FOA were used. The strains were incubated at 28 °C for 7 days, and all experiments were conducted in triplicate.

### 2.5. Acetamide Utilization Assay

*Spiromastix* sp. SCSIO F190 was tested for the ability to grow on 10 mM acetamide agar using a plate assay. Spores were inoculated onto plates using acetamide as the sole carbon and nitrogen source supplemented with different concentrations of cesium chloride. Plates were incubated at 28 °C for 14 days and then compared with the control plate containing both glucose and ammonium tartrate.

### 2.6. Protoplast Transformation

The protoplasts of *Aspergillus* sp. SCSIO SX7S7 and *Spiromastix* sp. SCSIO F190 were prepared as previously described [23]. In brief, the conidia were cultured in 50 mL potato dextrose broth (PDB) at 100 rpm and 28 °C until germination. After completion of the culture, the mycelia were collected and digested with Lysing enzymes (Sigma-Aldrich, St. Louis, MO, USA) and Yatalase (TaKaRa, Dalian, China). The digested protoplasts and mycelial debris were separated by gradient centrifugation. Finally, the protoplasts were resuspended in STC solution to a concentration around 2 × 10^7^ mL^−1^ for the subsequent transformation.

The transformation of exogenous plasmid was also performed according to our previously established method [23]. Briefly, 2–5 μg plasmid was added into the protoplast suspension, and the plasmid was introduced into the protoplast via PEG-mediated protoplast transformation. Next, the PEG-treated protoplasts were collected and poured onto 3–4 regeneration plates. ANM with 1.2 M sorbitol was used for preparing regeneration plates in the *pyrG* marker recycling system, while ANM with 1.2 M sorbitol but only containing acetamide as the sole carbon and nitrogen source was used in the *amdS* marker recycling system. After 4–20 days of cultivation at 28 °C, the transformants were inoculated onto a new PDA plate (without sorbitol), and the genome DNA of the transformants was extracted for genotyping PCR as described in our previous research [23].

### 2.7. Secondary Metabolite Analysis by HPLC

To examine the production of secondary metabolites, wild-type and mutants were cultivated in a PDB liquid medium for 7 days at 28 °C, 180 rpm. Then, the cultures were extracted with an equal volume of butanone by sonication, and the organic phases were evaporated to dryness. Next, the remaining residues were redissolved in 1 mL MeOH, and 50 μL of which was injected into the analytical HPLC for analysis. The HPLC analysis was performed on an Agilent 1260 HPLC system (Agilent Technologies Inc., Santa Clara, CA, USA) equipped with a binary pump and a diode array detector using an analytical Phenomenex column (250 × 4.60 mm, 5 microns). The injected samples were eluted with a linear gradient from 5% to 80% solvent B over 20 min, followed by 80% to 100% solvent B in 1.5 min, and then eluted with 100% solvent B in 5.5 min, at a flow rate of 1.0 mL/min using UV detection at 220 nm, 254 nm, 275 nm, and 354 nm.

## 3. Results and Discussion

### 3.1. Construction of CRISPR-Cas9 Vectors Containing pyrG/amdS Marker Gene

CRISPR−Cas9 genome editing technology plays an extremely significant role in synthetic biology and metabolic engineering [31]. In our previous study, a rearranged CRISPR-Cas9 vector containing *T. reesei* codon-optimized *cas9* (toCas9), *hph* coding sequences, and sgRNA expression cassette driven by RNA polymerase II promoters has shown to be an efficient gene editing tool in *Aspergillus* sp. SCSIO SX7S7 and *Spiromastix* sp. SCSIO F190 [23]. However, one marker gene is not sufficient to reveal the hidden BGCs in marine fungi using multi-round genetic engineering. Recyclable markers that can be removed from the targeted site after using can be valuable tools in ambitious metabolic engineering programs. The *pyrG* marker serves as a strong, recyclable, auxotrophic selection marker that can be used for uridine/uracil auxotrophic complementary transformation and then counter-selected using 5-FOA. The *pyrG* gene from *A. fumigatus* has been widely applied in filamentous fungi [29,32]. In addition to the *pyrG* marker, the *amdS* gene is another broadly used recyclable marker. In the fungi species lacking acetamidase activity, the expression of acetamidase gene from *A. nidulans* under a strong promoter provides selection on media containing acetamide as the sole nitrogen source that can be counter-selected using FAA [20,33,34]. The efficiency of looping out counterselectable markers can be drastically increased by employing the blaster strategy, flanking the marker with around 300 bp direct repeats (DRs) [16,35]. However, to our knowledge, this technique has not been employed in marine-derived fungi.

Thus, the *pyrG* sequence from *A. fumigatus* and the *amdS* gene from *A. nidulans* were selected to test their function in the marine-derived fungi. Since the promoter (P*trpC*) and terminator (T*trpC*) originated from *A. nidulans* are functioning well in *Aspergillus* sp. SCSIO SX7S7 and *Spiromastix* sp. SCSIO F190 [23], we kept the expression of *pyrG* and *amdS* under the control of P*trpC* and T*trpC*. Consequently, the sequence of *pyrG* or *amdS* along with P*trpC* and T*trpC* were ligated into the pBluescript II SK plasmid to generate marker expression vectors pBSKII-*pyrG* or pBSKII-*amdS*. Given the high efficiency of blaster counterselection methodology in removing counterselective markers, a 300 bp sequence located directly after the T*trpC* terminator in pBSKII-toCas9-*hph* plasmid was designed as a direct repeat (DR) sequence. The same 300 bp DR sequence was amplified for inserting before the P*trpC* promoter. Next, two new vectors called pBSKII-toCas9-*pyrG* and pBSKII-toCas9-*amdS*, respectively*,* were generated by inserting the marker expression cassette and the DR sequences into the *PsiI* digested pBSKII-toCas9-*hph* to replace the *hph* expression cassette (Figure 1). Finally, the amplified gene-specific sgRNA expression fragments were inserted into either *pyrG* or *amdS* vector via one step isothermal assembly to generate recyclable gene editing vectors, named pBSKII-toCas9-*pyrG*-sgRNA and pBSKII-toCas9-*amdS*-sgRNA (Figure 1).

Application of the *pyrG* selection marker is laborious in generating uracil/uridine auxotrophs but shows high gene editing efficiency due to the near-nominal level of false positive rate of transformed clones and the normal growth rate on a uracil/uridine depleted medium [10]. The *amdS* selection marker takes advantage of its ability to transform wild-type cells that lack a background auxotrophy [16]. However, the *amdS* selection marker can only be applied in the strains unable to grow (or with poor growth) on acetamide as the sole carbon- or nitrogen-source. Therefore, the simplicity and gene editing efficiency of these two systems would be species dependent. For example, our initial trial of the *amdS*-based recyclable system in *Aspergillus* sp. SCSIO SX7S7 failed (data not shown), which may be due to the high expression of endogenous acetamidase. Thus, it is important to establish a gene editing toolkit containing different recyclable markers, which will guarantee flexible application in marine-derived fungi with different and complicated genetic backgrounds.

### 3.2. Construction of Uracil Auxotroph Mutant by Disruption of the pyrG Gene in Aspergillus sp. SCSIO SX7S7

The first step for developing a *pyrG*-based marker recycling system is to generate a recipient strain with mutation of the *pyrG* gene for performing uracil auxotrophic complementary transformation and subsequent gene disruption. Most *pyrG* mutants were obtained following either ultraviolet (UV) or 5-FOA induced mutagenesis, which may include multiple undesirable mutations [12]. Moreover, the mutation rate of the *pyrG* gene under 5-FOA pressure is species dependent. In this study, the sensibility of *Aspergillus* sp. SCSIO SX7S7 to 5-FOA was tested, and we found that 1 mg/mL 5-FOA can completely inhibit the growth of SCSIO SX7S7 (Appendix A). We tried to induce spontaneous *pyrG* mutation in *Aspergillus* sp. SCSIO SX7S7 using the uridine/uracil-containing media supplemented with different concentrations of 5-FOA (1–10 mg/mL). We were unable to retrieve 5-FOA-resistant colonies. As our previously established CRISPR-Cas9 system has shown to be an efficient tool for gene disruption in *Aspergillus* sp. SCSIO SX7S7 [23], a CRISPR-Cas9 plasmid containing a sgRNA targeting the exon of *pyrG*-7S7 was generated and introduced into the protoplast of SCSIO SX7S7. After 4 days of growth, six colonies were picked, and four of them showed no growth on PDA but were able to grow on PDA supplemented with uridine/uracil. Moreover, the four colonies were 5-FOA resistant, suggesting that the four colonies may be *pyrG*-7S7 deficient (Figure 2A). PCR application using primers targeting *pyrG*-7S7 further validated the disruption of *pyrG*-7S7 gene (Figure 2C). The growth test revealed that disruption of *pyrG*-7S7 did not induce growth change compared with the wild type strain (Figure 2B), indicating the CRISPR-Cas9 system may not induce non-specific mutation. Therefore, one of the *pyrG*-7S7 mutants was selected for testing the feasibility of *pyrG-*based marker recycling system using the new plasmid built in this study.

### 3.3. Gene Disruption Using the CRISPR-Cas9-pyrG System in Aspergillus sp. SCSIO SX7S7

Our previous study showed that tricyclic depsidones and bicyclic depsides (DEPs) are constitutively produced by *Aspergillus* sp. SCSIO SX7S7 with high abundance [24]. To eliminate depsidones and depsides in *Aspergillus* sp. SCSIO SX7S7, the genome of SCSIO SX7S7 was analyzed by SM biosynthetic gene clusters annotation tool AntiSMASH [36], and a gene cluster containing two PKS genes (*depD* and *depH*) was proposed to be responsible for DEPs biosynthesis. Then, a sgRNA targeting the sequences of the PKS gene *depH* were designed and ligated into pBSKII-toCas9-*pyrG*. After transformation and selection on the plate without uridine/uracil, six transformants were randomly picked out for PCR validation using primers surrounding the sgRNA binding site. Our previous study has shown that gene disruption by the CRISPR-Cas9 system in the two tested marine-derived fungi can result in high frequency integration of the exogenous plasmid at the Cas9 cut site [23] and thus result in failed PCR when using primers flanking the PAM site. Similarly, five transformants harboring the *depH* disruption construct failed to be PCR amplified for obtaining targeted bands. (Appendix A). To assess whether the ∆*depH* mutant abolishes the ability to produce DEPs, the secondary metabolites from ∆*depH* mutant were extracted and analyzed with HPLC. HPLC analysis revealed that the ∆*depH* mutant was not able to produce DEPs (Figure 3), which further confirmed that the *depH* has been successfully mutated by the *pyrG*-based gene editing system and validated that the *depH* is a key gene for DEPs biosynthesis. In addition to DEPs, other differences were also observed in the metabolic profile between the wild type and ∆*depH* strains, suggesting that inactivation of DEPs biosynthesis cluster may also alter the biosynthesis of other NPs. Moreover, lack of visible growth differences between wild type and Δ*depH* mutant suggested that the editing happened at the loci of interest without non-specific disruptions (Figure 3).

### 3.4. Remove of pyrG Marker by Counter Selection

In the Δ*depH* mutant, all high-yielding DEPs in wild type strain were abolished, which provide us with a clean chassis cell for further natural product activation. To apply the chassis cell for the second round of genetic manipulation, we tried to remove the *pyrG* marker by counterselection. Due to the importance of DRs for looping out the *pyrG* marker [35] and the random break of transformed plasmid when integrated into the genome [23], PCR with the primers targeting the complete DRs franked *pyrG* marker expression cassette was carried out to check the integrity of DRs. All mutants retained the intact *pyrG* marker expression cassette flanked by DRs (Appendix A). One of the 7S7-Δ*depH-afpyrG* mutants was selected to remove the *pyrG* marker for yielding a marker-free mutant. The spores of the Δ*depH* mutant were spread on the 5-FOA (1 mM) plates, and the plates were incubated at 28 °C for 1 week until colonies became visible. Six colonies were picked out from the 5-FOA plate, and PCR amplifications with primers targeting the *pyrG* marker were used to confirm the deletion of the *pyrG* marker. As shown in Figure 4, selected colonies failed to be PCR amplified with the primers targeting *pyrG* marker, which depicted that the *pyrG* marker is absent from Δ*depH* mutants, and thus the marker-free Δ*depH* mutant (7S7-Δ*depH*) can be used for the stepwise gene manipulation for identifying novel natural products.

### 3.5. Gene Disruptions Using CRISPR-Cas9-amdS System in Spiromastix sp. SCSIO F190

The successful application of the *pyrG*-based CRISPR-Cas9 system in *Aspergillus* sp. SCSIO SX7S7 promoted us to use the same gene editing tool into *Spiromastix* sp. SCSIO F190. Unlike *Aspergillus* sp. SCSIO SX7S7, generation of the *pyrG* mutant in *Spiromastix* sp. SCSIO F190 failed after repeated attempts. This may be due to the high GC content of the four available sgRNAs targeting the *pyrG* gene in *Spiromastix* sp. SCSIO F190, which can reduce cleavage efficiency [37]. To improve the efficiency of the genetic engineering on the strains with complicated genetic backgrounds makes the construction of auxotrophic strains a laborious task. Thus, it is important to develop another marker recycling system that does not need to generate auxotrophic mutants.

The *amdS* gene encoding acetamidase from *A. nidulans* has been successfully used as a recycle marker in several filamentous fungi that are not able to use acetamide as the sole nitrogen or carbon source. To apply the *amdS* marker recycle system in *Spiromastix* sp. SCSIO F190, the metabolic capability of *Spiromastix* sp. SCSIO F190 for using acetamide was first evaluated. The spores of *Spiromastix* sp. SCSIO F190 were inoculated on the ANM plate with acetamide as the sole carbon and nitrogen source with different concentrations of cesium chloride. After 14 days of growth, *Spiromastix* sp. SCSIO F190 can grow weakly on the acetamide plate without cesium chloride, but the growth can be further inhibited by cesium chloride with increasing concentrations (Figure 5). The rationale for the effects of cesium chloride is not completely clear, but it has been used widely for suppressing the growth of recipient strains when the *amdS* marker recycle system was applied [20,38]. Therefore, the acetamide plate with 100 mM cesium chloride that can significantly block the growth of *Spiromastix* sp. SCSIO F190 was used to differentiate the transformed and untransformed cells.

Next, to evaluate whether the *amdS* marker from *A.nidulans* was suitable for gene knock-out in *Spiromastix* sp. SCSIO F190, a polyketide synthase gene (*spm11*) was selected for this proof-of-principle experiment. In our previous study, a gene cluster responsible for the biosynthesis of spiromarmycin, a high-abundance major product, was proposed, in which *spm11* was predicted as a backbone gene for spiromarmycin biosynthesis [25]. Thus, the absence of *spm11* would stop producing the brown spiromarmycin, giving a fast preliminary evaluation of targeted mutation by comparing the color difference on a solid medium. More importantly, clearance of the constantly produced major product is a key step for generating chassis cells with simplified HPLC background for exploring novel compounds. The pBSKII-toCas9-*amdS*-sgRNA plasmid containing a sgRNA targeting the *spm11* was transformed into *Spiromastix* sp. SCSIO F190 to inactivate the polyketide synthase gene. After transformation, cells were grown on the regeneration plates containing acetamide as the sole nitrogen source with 100 mM cesium chloride for 14 days, and the transformants were transferred into PDA plates. Among the ten picked colonies, eight of them showed different phenotypes with wild type strain (Appendix A).

The genome DNA of the eight colonies without significant accumulation of secreted compounds was extracted, and PCR was performed to check the mutation of the *spm11* gene. Unexpectedly, only one colony fail to obtain the PCR band when using the primers flanking the PAM site (Figure 6A). However, when the primers with one forward primer targeted the sgRNA binding site were used, all of the other seven colonies failed to obtain the PCR bands, suggesting the *amdS*-based CRISPR-Cas9 system-induced mutations are predominantly short deletions around the PAM site. Sequencing of the PCR bands of the seven colonies further confirmed that the *spm11* gene was disrupted by several DNA bases deletion/mutation (Figure 6B). In addition, HPLC analysis of a ∆*spm11* mutant showed that the mutant is not able to produce spiromarmycin (Figure 6C), validating the previous prediction that the *spm11* is a key gene for spiromarmycin biosynthesis.

### 3.6. Remove of amdS Marker by Counter Selection

To apply the resulting strain F190-∆*spm11*-*amdS* for another round of genetic manipulation, *amdS* maker excision aided by the DRs created upon integration under counter-selection pressure was performed. Similar to the *pyrG* marker looping out experiment, the integrity of DRs was first checked. Fortunately, six of the eight ∆*spm11* mutants containing the complete DRs franked *amdS* marker expression cassette (Appendix A). One of the six F190-∆*spm11*-*amdS* mutants was cultured on PDA plate for 10 days, and the spores were collected and spread on the 5 mM FAA plates. PCR analysis of the growing colonies on FAA plates confirmed correct marker removal (Figure 7). The resulting strain F190-∆*spm11* is ready for multiple rounds of genetic manipulation using an *amdS* maker, including inactivation of the other constitutively produced major products [26] and activation or heterogeneous expression of novel NPs in the chassis cells with simplified HPLC background.

## 4. Conclusions

In summary, we have successfully developed a genetic engineering toolbox containing two different marker recycling systems (*pyrG* and *amdS*) for gene disruption in marine-derived fungi. Applications of these marker recycling systems have proven to be efficient strategies for the clearance of the constitutively produced SMs in different marine fungi species. Especially, in *Spiromastix* species, it is the first application of the *amdS*-based recycle system for genome editing. Thus, the genetic editing systems established in this study showed promising potential in assisting sequential genetic manipulation of marine-derived fungi with distinct phylogenetics, providing powerful tools for accessing the structural diversity. Furthermore, although several model organisms such as *Aspergillus nidulans* and *Aspergillus oryzae* have been developed, there is no marine-derived model fungus. Here, with the assistance of recyclable markers, we are developing marine-derived fungi as model organisms/chassis cells for heterologous expressing BGCs originating from marine-derived fungi and investing marine fungi biology.

## Figures and Tables

**Figure 1 jof-09-00302-f001:**
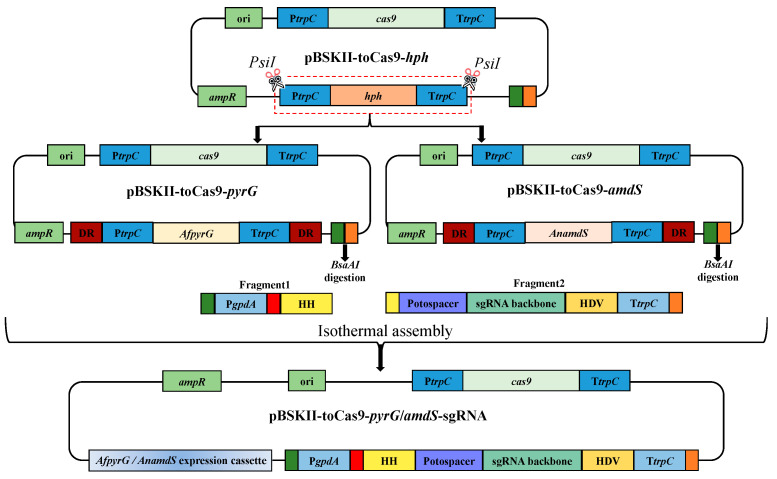
The construction of new CRISPR-Cas9 vectors (pBSKII-toCas9-*pyrG*/*amdS-*sgRNA) for multistep gene deletion in marine-derived fungi. The sequence of the *pyrG*/*amdS* expression cassette including P*trpC* promoter, T*trpC* terminator, and direct repeat was amplified. Then the amplified fragments were ligated into *PsiI* digested pBSKII-toCas9-*hph* vector using isothermal assembly to generate two vectors named pBSKII-toCas9-*pyrG* and pBSKII-toCas9-*amdS* which were further digested by the *BsaAI* restriction enzyme for building the final gene disruption construct. The Fragment1 and Fragment2 were amplified by Frag1 F/R and Frag2 F/R primers using FC330 plasmid as a template, respectively. The resultant PCR products of flanking regions and the *BsaAI*-digested pBSKII-toCas9- *pyrG*/*amdS* plasmid can be joined together to form the final construct pBSKII-toCas9-*pyrG/amdS*-sgRNA. For simplicity, all complementary ends are visualized in the same color and no DNA elements in the above figure are drawn to scale.

**Figure 2 jof-09-00302-f002:**
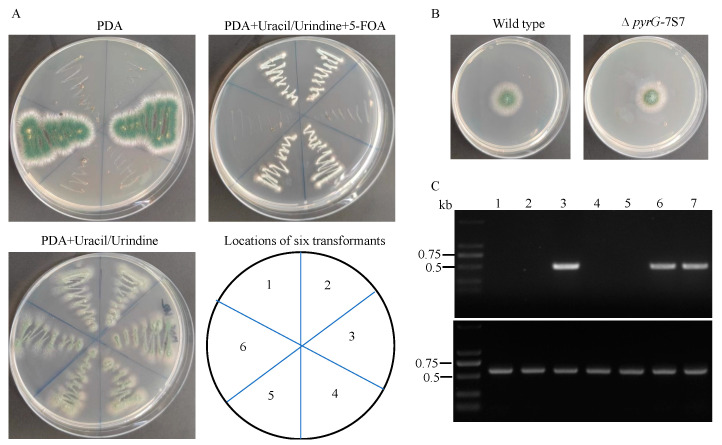
The inactivation of the *pyrG*-7S7 gene in *Aspergillus* sp. SCSIO SX7S7 using the CRISPR-Cas9 system. (**A**) Growth of the transformants carrying the CRISPR-Cas9 plasmids containing a sgRNA targeting exon of *pyrG*-7S7 on a PDA medium with or without uridine-uracil (5 mM) and a PDA medium supplemented with both uridine/uracil (5 mM) and 5-FOA (1 mg/mL). (**B**) Growth of the *Aspergillus* sp. SCSIO SX7S7 wild type and ∆ *pyrG*-7S7 strains on a PDA medium with uridine/uracil (5 mM) at 28 °C for 4 days. (**C**) DNA was extracted from the six randomly selected transformants (shown on the left side) and amplified by PCR using the primers flanking the PAM site (upper panel), and the DNA template quality of the tested samples was validated using primers flanking the ITS region (lower panel). PCR amplification with the same primers of DNA from the wild type strain served as positive control. (Lane 1–6: tested transformants; lane 7: wild-type strain).

**Figure 3 jof-09-00302-f003:**
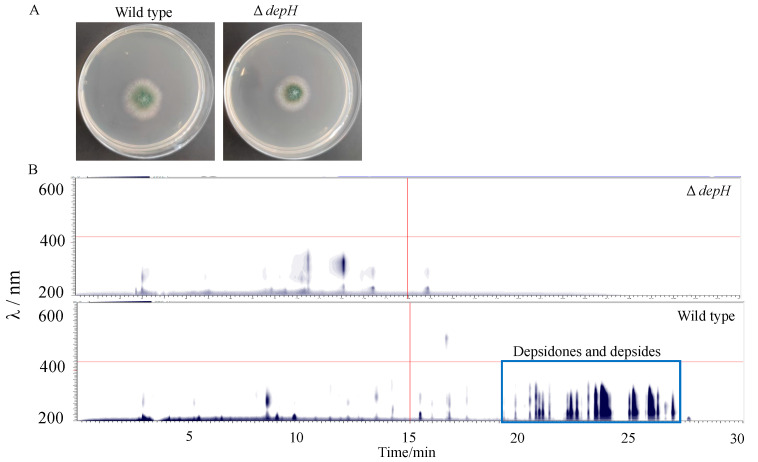
Characterization of the Δ*depH* mutant of *Aspergillus* sp. SCSIO SX7S7. (**A**) Growth of the wild-type and Δ*depH* mutation strains on PDA plates at 28 °C for 4 days. (**B**) HPLC-DAD contour plot of extracts from the wild-type and Δ*depH* mutation strains. The boxed region shows DEPs.

**Figure 4 jof-09-00302-f004:**
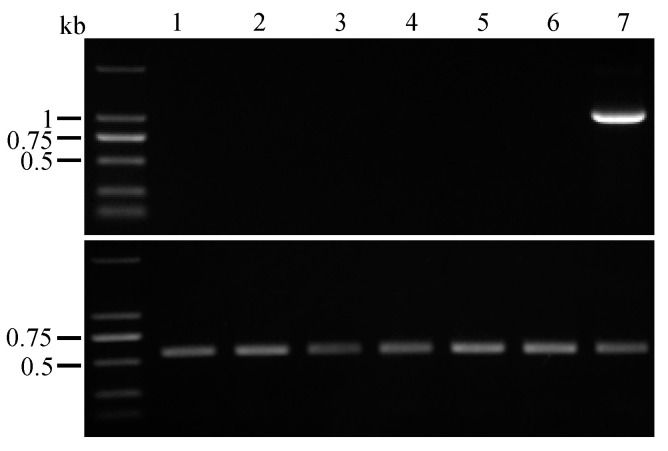
Verification of *AfpyrG* gene looping out. Upper panel: PCR was carried out using the primers targeting the *AfpyrG* marker. Lower panel: DNA template quality of the tested samples was validated using primers flanking the ITS region. The Δ*depH* strain carrying the *AfpyrG* marker was set as positive control. (Lane 1–6: Δ*depH* mutants without *AfpyrG* marker; Lane 7: positive control).

**Figure 5 jof-09-00302-f005:**
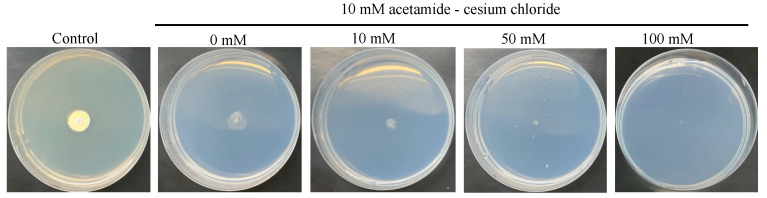
Growth of *Spiromastix* sp. SCSIO F190 on ANM plate with acetamide as sole nitrogen and carbon source. The spores of *Spiromastix* sp. SCSIO F190 were inoculated on the acetamide plate with different concentrations of cesium chloride and incubated at 28 °C for 10 days. The growth on the ANM plate containing glucose and ammonium tartrate was set as control.

**Figure 6 jof-09-00302-f006:**
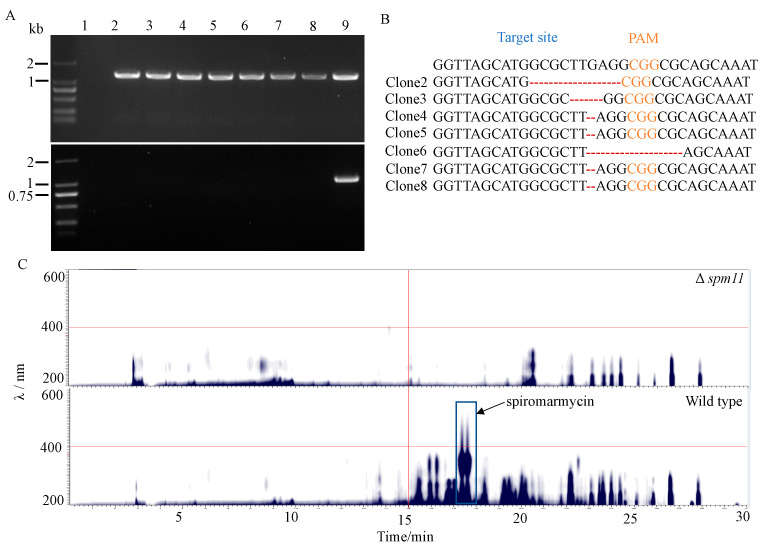
Inactivation of *spm11* in *Spiromastix* sp. SCSIO F190 using the *amdS*-based CRISPR-Cas9 system. (**A**) DNA amplification of the DNA region surrounding the sgRNA binding site of the eight clones with different phenotypes from wild type plate using primers flanking the PAM site in the *spm11* gene (Upper panel) and primers with one forward primer targeted the sgRNA binding site (Lower panel). PCR amplification with the same primers of DNA from the wild type strain served as positive control. (Lane 1–8: tested transformants; lane 9: wild-type strain.) (**B**) Sequence analysis of PCR products of clones 2–8. (**C**) HPLC-DAD contour plot of extracts from the wild-type and Δ*spm11* mutation strains. The boxed region indicates spiromarmycin.

**Figure 7 jof-09-00302-f007:**
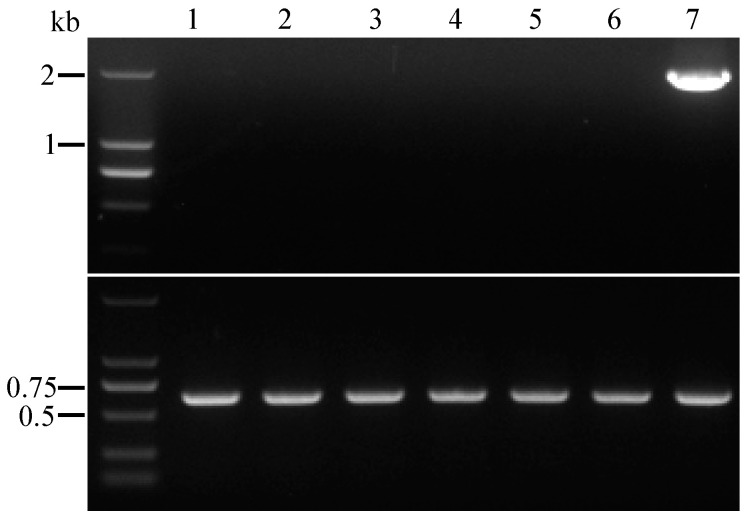
Verification of *AnamdS* gene looping out. Upper: PCR was carried out using the primers targeting the *AnamdS* marker. Lower: DNA template quality of the tested samples was validated using primers flanking the ITS region The F190-∆*spm11*-*AnamdS* strain carrying *AnamdS* marker was set as positive control. (Lane 1–6: Δ*spm11* mutants without *AnamdS* marker; Lane 7: positive control).

## Data Availability

The data presented in this study are available in this manuscript, and constructs can be requested from the corresponding author.

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
