# Peer review of "Development of Marker Recycling Systems for Sequential Genetic Manipulation in Marine-Derived Fungi Spiromastix sp. SCSIO F190 and Aspergillus sp. SCSIO SX7S7"

_jof, 2023, doi:10.3390/jof9030302_

Round 1

Reviewer 1 Report

The paper by Chan et al., presents two new tools designed with a combination of recyclable selective markers and CRISPR/Cas gene editing in order to disrupt genes that lead in high yield of specific secondary metabolites. This engineering could allow novel (silent) secondary compounds to be identified/expressed in marine fungi (or other fungi with less known genetic information). These editing tools are always appreciated and important in secondary metabolite research. The work done in this paper is laborious and contains different “omic” (antiSMASH), analytical (HPLC) and molecular bench (design of constructs, culturing efforts etc) techniques. This reviewer has no objection for the paper to be published in JoF, but this is a high-tier journal so significant corrections in language are mandatory. Parts of the text are easy to follow while others require efforts in order to be understood. Some edits and corrections are provided below. Also, an overall comment is that the authors need to make sure that they stress out the caveats of their experimental design. As an example, the authors conclude: “in assisting sequential genetic manipulation of marine derived fungi with distinct phylogenetics,”. Nowhere in the text authors discuss or point out phylogenetic differences between the strains used. The authors point the reader to previous publications (which is fine), but since they do not provide evidence in this paper on phylogenetics (no phylogenetic tree is available in the manuscript, no citations on how similar or different are those strains used) authors cannot jump into such broad and without showning evidences in their text. Another example is that the authors inhibited the production of DEPs (and found the key genes for their production; all these are excellent). Nonetheless, their overall HPLC chromatograms show differences in other regions between mutant and wild type, these differences and what may signify are not commented in the text. In some cases, the differences are in favor of the wild type (more tense and abundant peaks compared to mutant) but sometimes also mutant shows more tense peaks vs. the wild type.  Albeit, the suppression of DEPs is discussed, the possibility that this suppression can also lead to suppression of silent NPs is not mentioned. Indeed, high yield of specific secondary metabolites can mask silent metabolites, but it seems that in some cases deleting the secondary metabolite BGCs with high yields might also lead to no metabolites at all. In this manuscript antiSMASH was used to justify the selection of the specific BGCs targeted (disrupted). Did the authors run antiSMASH to the mutant(s) after all the manipulations were performed in order to see what changed in the secondary metabolite landscape of the fungal strain after this extensive editing? (this applies to both Aspergillus and Spiromastix). Finally, the fungal strains used in this study are marine. Nonetheless, the media used did not contain sea salts (at least from what this reviewer understood). This can be a factor that dictates what metabolites are expressed vs. what can be silent. These types of caveats, is what this reviewer thinks that need to be interpreted in various parts of the text (like a sentence or two).  

L18: available selectable markers: available selection markers

L19: in genetic engineering programs for awaking: in genetic engineering for expressing silent

L22: “a relatively clean HPLC background was built by inactivating a polyketide synthase. Please rephrase; relative clean to what, and what was the aim of the HPLC analyses? This reviewer understands what you mean, but the phrasing is off…

Also, please change everywhere in the text: “Marine-derive fungi” to “Marine fungi”.

L37: points this group of microorganisms: shows that this group of microeukaryotes

L38: Unfortunately: Nonetheless,

L40: under standard laboratory conditions: I would suggest to refrain using “standard” laboratory conditions, since “standard” conditions differ between fungal strains/taxa. So, either define what is considered “standard” for your study, or rephrase as follows: under laboratory conditions.

L40: With the advent of genome sequencing and mining technologies, a variety of genome-mining methods have been developed to explore new natural products whose BGCs are ordinarily silent.: The plethora of advanced tools for genome sequencing and mining enabled the discovery of new secondary metabolites which are “silent” and are not expressed under laboratory conditions.

L45: Additionally: Likewise,

L47: Irrespective of the strategy employed: Despite of the strategy used,

L52: Clean chassis cells : pure chassis (self-replicating) cells,

L59: exogenous uridine/uracil: exogenous uridine/uracil (uridine/uracil auxotrophs)

L67: Nevertheless, there still have shortcomings in employing the pyrG-based marker recycling system, such as it may be laborious and time-consuming to generate pyrG mutants of some marine-derived fungi for which no genetic tools are available or the genetic background is not well understood.: Nevertheless, when applied to marine fungal strains with limited available sequencing and genetic information, the pyrG recycling system can be a laborious, time-consuming and inconclusive process.  

L80: Despite the utility of pyrG or amdS as a powerful recycling marker that greatly facilitates the genetic manipulation of various fungi species, the applications of these two counterselectable markers in marine-derived fungi are still very limited: Despite the successful use of pyrG and/or amdS as recycling makers in various fungal taxa, these two counterselectable markers are seldomly utilized in genetic manipulation of marine fungi.

L84: Several antimicrobial natural products have been identified from these two strains with high yielding: Several antimicrobial natural products (e.g., XX, ZZ) with high yield, have been identified from these two strains.

L87: be further activated by genetic approaches. : be further activated in order to be expressed.

L91: Stopping: inhibiting the production of major secondary metabolites (e.g., XXX)

L98: In this work: In this study,

L101: Two newly built vectors: The two new constructed vectors,

L106: Again: with a relatively simple HPLC: These are things that are vaguely written... Relative and simple relative to what?

L111: for awakening silent NPs.: that led to expression of silent NPs.

Overall in Material and Methods you need to be precise and cite more often your supplementary tables.

L119: Acetamide: acetamide (provide % or concentration). It is easier for the reader to track exactly what you did instead of digging in to the references you provide.

L123: with appropriate antibiotics: name the antibiotic instead of appropriate antibiotic; with XXX (ZZmg/ml)

L136: out with commercial kits (Omega 136 Bio-Tech, Inc., Norcorss, GA, USA): out with commercial kits (Omega 136 Bio-Tech, Inc., Norcorss, GA, USA), following the manufacturer’s protocol. (same for other places)

L217: However, only one drug resistance marker is not sufficient for un- earthing the hidden biosynthetic potential in marine-derived fungi by multi-round genetic engineering. However, one marker gene is not sufficient to reveal hidden BGCs in marine fungi using multi-round genetic engineering.

L220: are therefore: can be, therefore,

L245: Based on the above information, the: Thus,

L260: pBSKII-toCas9-pyrG-sgRNA 260 and pBSKII-toCas9-amdS-sgRNA: pBSKII-toCas9-pyrG-sgRNA 260 and pBSKII-toCas9-amdS-sgRNA, respectively

L262: Either of these two recycling systems shows its own strengths and weakness: Please delete sentence.

L262: For the pyrG marker, although it’s time consuming to generate uracil/uridine auxotroph mutant it shows high efficiency of gene editing owing to the near-nominal level of false positive rate of transformed clones and normal growth rate on uracil/uridine depleted medium: pyrG selection marker is laborious in generating uracil/uridine auxotrophs, but shows high gene editing efficiency due to the near-nominal level of false positive rate of transformed clones, and the normal growth rate on uracil/uridine depleted medium.

L266-268: While the amdS marker takes advantage of its ability to transform wild-type cells that lack a background auxotrophy: While amdS selection markers transform successfully wild-type cells that lack auxotrophy, these selection markers can only be applied in fungal strains unable to grow (or with poor growth) on acetamide as sole carbon or nitrogen source.

L270: was failed: failed

L285: Then, we tried to induce spontaneous pyrG mutation in Aspergillus sp. SCSIO SX7S7 using the uridine/uracil-containing media supplemented with different concentrations of 5-FOA (1-10 mg/mL), but we didn’t get any 5-FOA-resistant colony. We induced spontaneous pyrG mutation in Aspergillus sp. SCSIO SX7S7 using the uridine/uracil-containing media supplemented with different concentrations of 5-FOA (1-10 mg/mL). We were unable to retrieve 5-FOA-resistant colonies. (also the “then,” in other places needs to be rephrased)

L316: In previous chemical investigations, we found that tricyclic depsidones and bicyclic depsides (DEPs) are constitutively produced by Aspergillus sp. SCSIO SX7S7 with high 317 abundance: Our previous studies showed that that tricyclic depsidones and bicyclic depsides (DEPs) are constitutively produced by Aspergillus sp. SCSIO SX7S7 and at high abundances (XXX mg/ml).

L330: it is not able :  was not able to produce DEPs (Figure 3). Absence of DEPs confirmed that depH has been successfully mutated by the pyrG-based gene editing system, and validated that depH is key gene in DEPs biosynthesis.

L333: Moreover, there are no visible growth differences between wild type and depH mutation strains, suggesting non-specific gene disruption may not happen in other genome regions (Figure 3): Lack of visible growth differences between wild type and depH mutants, suggested that the editing happened at the loci of interest without non-specific disruptions. Nonetheless, this needs to be further confirmed since non-specific gene disruptions does not always produces phenotypes.   

L343: all high-yielding DEPs in wild type strain disappeared: the expression of DEPs was inhibited

L349: Luckily,: delete luckily.

L350: Then: delete then, and start sentence as One,

L356: to get the PCR bands: to be PCR amplified

L356: As shown in Figure 4, all selected colonies failed to get the PCR bands shown in the ΔdepH mutant containing pyrG marker expression cassette(Figure 4), which demonstrated that the pyrG marker in the ΔdepH mutant had been evicted, and thus the marker-free ΔdepH mutant called 7S7-â–³depH is ready for the stepwise gene manipulation for excavating novel natural products: Selected colonies failed to PCR amplify, which depicts that pyrG marker is absent from ΔdepH mutants (7S7-â–³depH), and can be now used for stepwise gene manipulation for identifying novel natural products.

L369: Due to the feasibility of the pyrG-based CRISPR-Cas9 system in Aspergillus sp. SCSIO SX7S7, we tried to apply this system in Spiromastix sp. : The successful application of pyrG-based CRISPR-Cas9 in Aspergillus sp. SCSIO SX7S7 prompted us to use the same gene editing tool also into in Spiromastix sp. SCSIO F190.

L372: This may be due to the four available sgRNAs targeting the pyrG gene in Spiromastix sp. SCSIO F190 are high GC content, which can lead to reduced cleavage efficiency. This may due to the high GC content of the four available sgRNAs targeting the pyrG gene in Spiromastix sp. SCSIO F190 which can to reduce cleavage efficiency.       

Reviewer 2 Report

The manuscript by Chen et al. describes the development of marker recycling systems for two marine-derived fungi, Aspergillus and Spiromastix. These tools almost certainly will be very valuable in further genetic engineering. The data are well presented and documented.

My only point of criticism is with regard of figure 5 which in my opinion is poorly done and does not follow standard microbial techniques. Here a dilution streaking would have been required (e.g. https://en.wikipedia.org/wiki/Streaking_(microbiology))

this should be changed prior to publication.

Round 2

Reviewer 1 Report

The authors performed appropriate corrections. This reviewer endorses their work for publication. Nonetheless, please make sure at the proofreading stage for grammatical errors that can escape.

As a small comment correct capital L to lower l:

 Moreover, Lack of visible growth differences between wild type and depH mutants, suggested that the editing happened at the loci of interest without non-specific disruptions

Author Response

Point 1: The authors performed appropriate corrections. This reviewer endorses their work for publication. Nonetheless, please make sure at the proofreading stage for grammatical errors that can escape.

As a small comment correct capital L to lower l:

Moreover, Lack of visible growth differences between wild type and depH mutants, suggested that the editing happened at the loci of interest without non-specific disruptions.

Response 1: Thank you for your careful review. We have now revised capital L to lower l in Line 339. In addition, we have carefully checked the manuscript again and several spelling or grammatical errors have been revised (Kindly see lines 22, 80, 139, 332, 414, and 455, marked in red).